

# High circulating fibroblast growth factor-21 levels as a screening marker in fatty pancreas patients

Fei Han[1,2,*], Ling Yin[2,*], Xiaoping Yu[3,4], Renyan Xu[3,4], Mingxiang Tian[3], Xinnong Liu[5], Lu Zhou[2], Lianghao Hu[6], Weijuan Gong[2,7], Weiming Xiao[2], Guotao Lu[2], Guanghuai Yao[2] and Yanbing Ding[1,2]

[1] Dalian Medical University, Dalian, China
[2] Department of Gastroenterology, Affiliated Hospital of Yangzhou University, Yangzhou, China
[3] Department of Health Promotion Center, Affiliated Hospital of Yangzhou University, Yangzhou, China
[4] Department of Ultrasound, Affiliated Hospital of Yangzhou University, Yangzhou, China
[5] Institute of Digestive Diseases, Yangzhou University, Yangzhou, China
[6] Department of Gastroenterology, Changhai Hospital, Second Military Medical University, Shanghai, China
[7] Jiangsu Co-innovation Center for Prevention and Control of Important Animal Infectious Diseases and Zoonoses, College of Veterinary Medicine, Yangzhou University, Yangzhou, China
[*] These authors contributed equally to this work.

Corresponding authors
Guanghuai Yao, ghyao@yzu.edu.cn
Yanbing Ding, ybding@yzu.edu.cn

## ABSTRACT

**Background.** The study aimed to detect the serum levels of fibroblast growth factor-21 (FGF-21) in fatty pancreas (FP) patients and to investigate their potential clinical value.
**Methods.** We screened patients with FP using transabdominal ultrasound. The anthropometric, biochemical and serum levels of FGF-21 were compared between the FP group and the normal control (NC) group. A receiver operating characteristic (ROC) curve was used to evaluate the predictive value of serum FGF-21 for FP patients.
**Results.** Compared with the NC group, body mass index, fasting blood glucose levels, uric acid levels and cholesterol levels of the FP group were significantly higher, while the high-density lipoprotein level was lower. In addition, levels of serum FGF-21, resistin, leptin and tumor necrosis factor-$\alpha$ were significantly higher than those in the NC group, while the serum adiponectin level was lower. Pearson analysis showed serum FGF-21 levels in FP patients were negatively correlated with leptin. The ROC curve showed the best critical value of the serum FGF-21 level in FP patients was 171 pg/mL (AUC 0.744, $P = 0.002$, 95% confidence intervals 0.636–0.852).
**Conclusion.** Serum FGF-21 was closely related to fatty pancreas. Detecting serum FGF-21 levels may help identify the population susceptible to FP.

## INTRODUCTION

Adipose tissue, as the energy storage of the body, is mainly distributed throughout the body subcutaneously and around the abdominal organs. An imbalance in energy metabolism can lead to fat deposition in abnormal areas, such as the liver and muscle fascia, and these

abnormal deposits are called ectopic fat deposits (*van Herpen & Schrauwen-Hinderling, 2008*). It has been found that fat can be deposited heterotopically in pancreatic tissue, which is usually referred to as fatty pancreas (FP) and this condition has a definite pathophysiological significance (*Della et al., 2015*). As early as 1933, *Ogilvie (1933)* found via autopsy that there was a difference in pancreatic fat content between obese and nonobese people. *Li et al. (2017)* noted that the incidence of fatty pancreas varied from 16% to 69.7% in different countries. Previously, our team found that the prevalence of fatty pancreas in the examined population was approximately 2.7% in Yangzhou, China (*Wang et al., 2018*). The specific pathogenesis of fatty pancreas remains unclear.

Fibroblast growth factor-21 (FGF-21) is a member of the endocrine fibroblast growth factor subfamily (*Fisher & Maratos-Flier, 2016*). The main physiological function of FGF-21 is to maintain the balance between glucose and lipid metabolism (*Fisher & Maratos-Flier, 2016*). Animal experiments have shown that excessive expression of FGF-21 could lead to body fat reduction (*Micanovic et al., 2009*). In addition, exogenous FGF-21 plays a clear role in promoting glucose uptake and reducing blood glucose and liver fat deposition (*Micanovic et al., 2009*). Clinical studies have shown that the circulating FGF-21 level was closely related to metabolic diseases. Serum FGF-21 levels in patients with fatty liver, obesity, metabolic syndrome and diabetes were significantly higher than those in healthy volunteers (*Li et al., 2018*; *Akour et al., 2017*; *Li et al., 2010*; *Keuper, Häring & Staiger, 2019*).

FGF-21 also plays an important role in pancreatic diseases (*Coate et al., 2017*; *Hernandez et al., 2020*; *Wang et al., 2019*). FGF-21 is highly expressed in pancreatic exocrine cells, and its expression can be significantly increased when acute pancreatitis occurs (*Fon et al., 2010*). *Johnson et al. (2014)* showed that FGF-21 gene knockout mice experienced aggravated acute pancreatitis and that exogenous FGF-21 could significantly reduce the severity of caerulein-induced acute pancreatitis in mice. Recent animal experiments have confirmed that recombinant human FGF-21 can reduce inflammation, pancreatic cysts high-grade intraepithelial neoplasia and pancreatic cancer in high-fat-diet-fed mice, suggesting that FGF-21 may be used for the prevention and treatment of pancreatic cancer (*Luo et al., 2019*). Surprisingly, FGF-21-deficient mice exhibit a phenotype of pancreatic fat deposition (*Johnson et al., 2009*). However, there are no reports on the correlation between FGF-21 and fatty pancreas. Hence, in the present study, circulating FGF-21 levels in patients with fatty pancreas were detected for the preliminary assessment of their relationship.

## MATERIAL AND METHODS

### Study population

The study was performed at the Affiliated Hospital of Yangzhou University in Yangzhou, China. We selected 99 fatty pancreas subjects and 16 healthy normal control subjects from the physical examination center of Affiliated Hospital of Yangzhou University from August 2018 to June 2019. Fatty pancreas subjects who met the following criteria were excluded: (1) subjects aged <18 years or >65 years; (2) subjects with acute or chronic inflammatory diseases; (3) subjects with a previous diagnosis of chronic pancreatic, liver

or kidney disease; (4) subjects with severe immune system disorders or pregnancy; (5) subjects with incomplete information or refusal to provide clinical blood samples. Besides, the inclusion criteria of normal control subjects were as follows: (1) subjects aged 18–65 years; (2) subjects without fatty pancreas; (3) subjects without any acute or chronic diseases; (4) subjects without previous histories of acute or chronic diseases; (5) subjects without severe immune system disorders or pregnancy or cancers; (6) subjects without incomplete information or refusal to provide clinical blood samples. The exclusion criteria were those who did not meet the inclusion criteria.

This study conformed to the ethical principles of the Declaration of Helsinki. The study was approved by Ethics Committee of Affiliated Hospital of Yangzhou University (Ethical Application Ref: 2018-YKL11-27- topic1). All participants agreed to participate in the study, and written informed consent was obtained from each subject.

## Diagnosis of fatty pancreas and fatty liver

As previously described, all subjects underwent transabdominal ultrasonography to diagnose fatty pancreas and fatty liver (*Wang et al., 2018*). Operations were performed by skilled surgeons with more than 10 years of experience using transabdominal ultrasonography (with an abdominal convex array probe, frequency: 3.5–5 MHz, LOGIQ E9, GE, USA). The ultrasound diagnostic criteria for fatty liver were as follows: the anterior echo of the liver was enhanced while the posterior echo was weakened, and the tubular structure of the liver could not be clearly displayed (*Hamaguchi et al., 2007*). The characteristics of the ultrasonic image of fatty pancreas were as follows: diffuse strong echoes of pancreatic parenchyma, normal or slightly increased volume, similar or slightly higher echogenicity compared to the adipose tissue in the area of the superior mesenteric artery (*Wang et al., 2018*; *Smereczynski & Kołaczyk, 2016*).

## Anthropometric and biochemical findings

Clinical information, including the subject's identity, age, gender, body weight, height, blood pressure, past medical history, drug history, history of smoking and alcohol intake, was recorded using a standardized questionnaire. Height and body weight were assessed using standardized and calibrated scales. Additionally, the body mass index (BMI) was calculated for each subject (BMI = body weight (kg)/square of height ($m^2$)). The subjects included "continuous smokers" (continuous or cumulative smoking for 6 months or more in a lifetime) and "nonsmokers" (including those who quit smoking ≥1 year ago) (*Wang et al., 2018*). The subjects also included "continuous drinkers" (drinking volume ≥20 g/d; drinking duration ≥2 years) and "nondrinkers" (individuals who had not consumed alcohol for ≥half a year) (*Wang et al., 2018*).

All subjects fasted for at least 8 h the night before the visit, and blood samples were collected with the participants in a quiet state on an empty stomach the next morning. Blood samples were sent to the laboratory for uniform testing (dry chemical method), and the remaining unused serum was frozen in an ultralow temperature freezer for analysis using an enzyme linked immunosorbent assay (ELISA). Diabetes was defined as a fasting blood glucose (FBG) level ≥ 7.0 mmol/L or a previous diagnosis by a doctor (*Jia et al., 2019*).

Dyslipidemia was defined as meeting any of the following: high serum total cholesterol (CHO) ($\geq$5.17 mmol/L), high triglyceride (TG) levels ($\geq$1.7 mmol/L), decreased high density lipoprotein (HDL) levels (<1.03 mmol/L), high low density lipoprotein (LDL) levels ($\geq$4.1 mmol/L), or a previous diagnosis of dyslipidemia by a doctor (*Wang et al., 2018*).

## Measurement of FGF-21 and adipocytokine levels in human serum

Levels of serum FGF-21, adiponectin, leptin, resistin and tumor necrosis factor-alpha (TNF-$\alpha$) were quantified using ELISA kits (USCN Kit Inc., Wuhan, China). All operations were carried out in strict accordance with the kit instructions. The quantification of the results was calculated according to the instructions. Averaged the readings for standards, controls, and samples and subtracted the average zero standard relative light unit. Created the standard curve on log–log graph paper, with FGF-21 or adipocytokines concentration on the $y$-axis and the relative light unit value on the $x$-axis. Drawn the best fit straight line through the standard points and it could be determined by regression analysis. If samples had been diluted, the concentration read from the standard curve must be multiplied by the dilution factor.

## Statistical analysis

Statistical analysis was performed with IBM SPSS 19.0 software. Normality and homogeneity of variance tests were performed for each group of data by Shapiro–Wilk and Levene's test. Continuous measurement data that were normally distributed are presented as the mean $\pm$ standard deviation (Mean $\pm$ SD). Continuous variables that were not normally distributed are presented as medians (25th and 75th percentiles), and categorical variables are presented as percentages (n%). The differences between the fatty pancreas (FP group) and normal control (NC group) groups were determined using Student's t tests and Mann–Whitney U tests. The qualitative data were compared using the chi-square test. Pearson analysis was used to evaluate the correlation between FGF-21 and other factors. GraphPad Prism 7 was used to generate receiver operating characteristic (ROC) curve. A two-sided $p$ value <0.05 was used to indicate statistically significant differences.

# RESULTS

## Basic and clinical characteristics of the study population

In this study, 99 fatty pancreas patients (FP group) were enrolled, and 16 healthy subjects were enrolled as the normal control group (NC group). As shown in Table 1, the proportion of males in the FP group and NC group was 58.6% (58/99) and 50% (8/16), respectively ($P = 0.524$). The average age and BMI in the FP group were significantly higher than those in the NC group ($P = 0.003$, $P < 0.001$). There was no difference in the proportion of smokers or drinkers between the two groups. Besides, the proportion of fatty liver, diabetes, hypertension, obesity, dyslipidemia in FP group were significantly increase than that in NC group (All $P < 0.001$). This also suggested that fatty pancreas may have an inseparable relationship with metabolic diseases.

Then, clinical characteristics were compared between the FP and NC groups. As shown in Table 2, it is self-evident that some key metabolic indicators, including serum uric acid

**Table 1  Demographic characteristics of participants.**

| Characteristic | FP( $N = 99$) | NC( $N = 16$) | P value |
|---|---|---|---|
| Males, (n%) | 58(58.6%) | 9(50%) | 0.524 |
| Age, y | 48.0 ± 11.4 | 38.6 ± 12.4 | 0.003[**] |
| BMI, kg/m$^2$ | 25.4 ± 3.2 | 21.7 ± 1.7 | <0.001[***] |
| Smoking | 18(18.2%) | 3(18.8%) | 0.957 |
| Drinking | 21(21.2%) | 4(25%) | 0.736 |
| Fatty liver | 25(25.3%) | 0 | <0.001[***] |
| Diabetes | 24(24.2%) | 0 | <0.001[***] |
| Hypertension | 28(28.3%) | 0 | <0.001[***] |
| Obesity | 14(14.1%) | 0 | <0.001[***] |
| Dyslipidemia | 50(50.5%) | 0 | <0.001[***] |

Notes.

Values are expressed in the mean ±SD or n%.

FP, fatty pancreas; NC, normal control; BMI, body mass index; SD, standard deviation.

[**]$P < 0.01$

[***]$P < 0.001$

(UA), glucose, CHO and TG were significantly higher in the FP group than in the NC group (All $P < 0.05$), with the corresponding serum HDL were decreased ($P < 0.001$).

## Comparison of serum FGF-21 and adipocytokine levels

We measured the levels of serum FGF-21 and adipocytokines (adiponectin, resistin, leptin and TNF-α) in the two groups. As shown in Fig. 1, the serum levels of FGF-21 and resistin in the FP group were significantly higher than that in the NC group, and adiponectin showed the opposite trend (All $P < 0.05$). In addition, the serum leptin and TNF-α levels in the FP group showed the upward trend in the FP group without a significant difference. Interestingly, in Table S1, we found that the levels of serum FGF-21 in male of FP group were significantly higher than that in female of FP group ($P = 0.004$). However, the levels of serum leptin and TNF-α in female of FP group were significantly higher than that in male (All $P < 0.01$). There was no significant difference between the two groups in serum resistin and adiponectin levels (Table S1).

It is remarkable that serum FGF-21 levels were negatively correlated with leptin ($r = -0.261$, $P = 0.005$) in the Pearson correlation analysis. In addition, there was no clear correlation with adipocytokines (adiponectin, resistin and TNF-α) or key metabolic clinical indicators (Glucose, CHO, LDL, TG and HDL) (Table 3).

## Fatty pancreas prediction analysis

Serum FGF-21 levels showed significantly greater discriminative ability for FP. Area under ROC curves (AUC) of FGF-21 were remarkably larger than adipocytokines (adiponectin, resistin, leptin and TNF-α) (Fig. 2). To determine the critical value of FGF-21 for predicting FP, ROC curves were assessed and are presented in Table 4. The results showed that 171 pg/mL was the best critical value (AUC 0.744, $P = 0.002$, 95% Confidence Interval (95% CI) 0.636−0.852). The corresponding sensitivity was 80.8%, and the specificity was 68.8% (Table 4).

**Table 2** Biochemistry characteristics of participants.

| Characteristic | FP( $N = 99$ ) | NC( $N = 16$ ) | $P$ value |
|---|---|---|---|
| TP, g/L | $73.2 \pm 4.7$ | $72.5 \pm 3.5$ | 0.556 |
| ALB, g/L | $44.3 \pm 2.1$ | $44.6 \pm 2.9$ | 0.638 |
| TB, umol/L | 13.3(10.9,16.3) | 16.0(12.4,18.9) | 0.056 |
| DB, umol/L | $3.7 \pm 1.2$ | $4.2 \pm 1.8$ | 0.282 |
| ALT, U/L | 21.0(15.2,33.0) | 24.5(15.3,33.8) | 0.982 |
| AST, U/L | 20.4(17.0,25.2) | 20.6(16.1,25.7) | 0.332 |
| $\gamma$-GGT, U/L | 27.0(18.6,39.6) | 15.3(12.6,22.7) | 0.190 |
| Cr, umol/L | $70.8 \pm 16.2$ | $70.0 \pm 12.6$ | 0. 847 |
| BUN, mmol/L | $5.2 \pm 1.3$ | $5.0 \pm 1.5$ | 0.608 |
| UA, umol/L | 337.0(286.0,404.3) | 288.0(250.8,319.0) | 0.004[**] |
| GLU, mmol/L | 5.5(5.1,6.1) | 5.0(4.7,5.1) | <0.001[***] |
| TG, mmol/L | $2.6 \pm 2.4$ | $1.3 \pm 0.4$ | <0.001[***] |
| CHO, mmol/L | $4.8 \pm 1.0$ | $4.3 \pm 0.6$ | 0.034[*] |
| HDL, mmol/L | $1.2 \pm 0.3$ | $1.4 \pm 0.3$ | <0.001[***] |
| LDL, mmol/L | $2.6 \pm 0.8$ | $2.4 \pm 0.4$ | 0.254 |

**Notes.**

Values are expressed in the mean $\pm$SD or the medians (25th and 75th percentiles).

FP, fatty pancreas; NC, normal control; TP, total protein; ALB, Albumin; TB, total bilirubin; DB, direct bilirubin; ALT, alanine aminotransferase; AST, aspartate aminotransferase; $\gamma$-GGT, gamma-glutamyl transpeptidase; Cr, creatinine; BUN, blood urea nitrogen; UA, uric acid; GLU, glucose; TG, triglyceride; CHO, cholesterol; HDL, high density lipoprotein; LDL, low density lipoprotein; SD, standard deviation.

[*]$P < 0.05$

[**]$P < 0.01$

[***]$P < 0.001$

# DISCUSSION

Although FP was discovered many years ago, its pathophysiology was still unclear. Most of the existing studies focused on clinical morbidity and its related risk factors (*Lesmana et al., 2015*; *Pham et al., 2016*). Previous studies showed that FP was closely related to metabolic diseases such as dyslipidemia, diabetes and fatty liver (*Wang et al., 2014*; *Makino et al., 2016*; *Hales et al., 2007*). Increased age, central obesity, and fatty liver were independent risk factors for FP (*Wang et al., 2018*). To date, there are no clinical molecular indicators for predicting or diagnosing FP.

FGF-21 is a recently discovered cytokine closely related to glycolipid metabolism and is the only protein in the fibroblast growth factor family that has no mitogenic activity (*Kharitonenkov et al., 2005*). The physiological function of FGF-21 is mainly involved in glycolipid metabolism and the metabolic regulation of insulin, body weight reduction and insulin resistance improvement (*Zhang et al., 2008*; *Chavez et al., 2009*; *Zhang & Li, 2015*). High-fat-diet-fed FGF-21-deficient mice exhibited liver fat accumulation and obvious lipid metabolic disorders; in contrast, injection of FGF-21 protein into diet-induced diabetic mice reversed the steatosis of the liver and restored the normal structure of the liver (*Murata, Konishi & Itoh, 2011*; *Xu et al., 2009*). Additionally, serum FGF-21 levels in fatty liver or diabetic patients were higher than those in the participants in the NC group, with a negative correlation with serum HDL and a positive

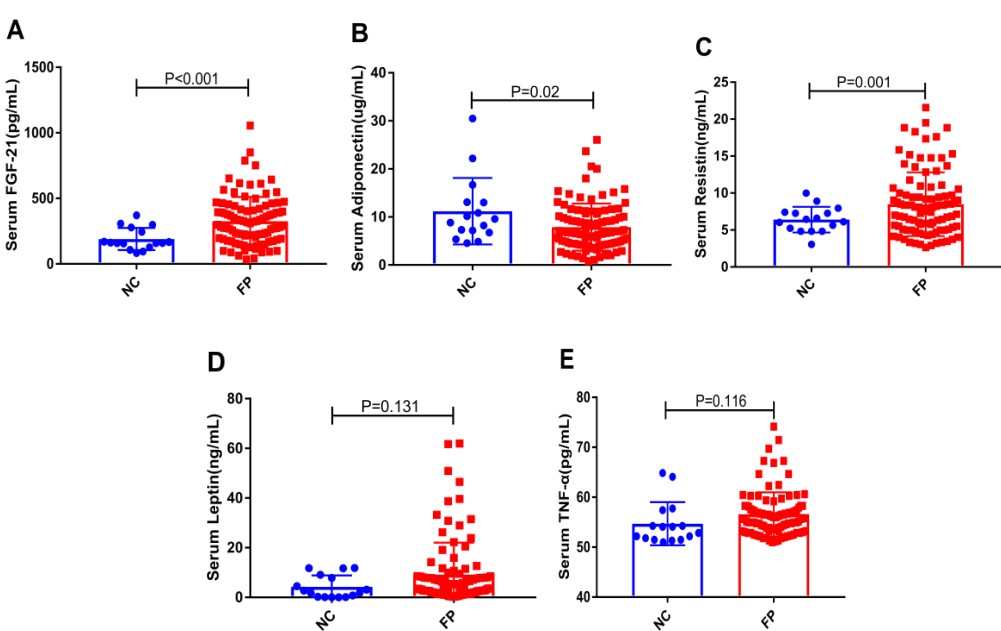

**Figure 1** (A–E) Comparisons of the levels of serum FGF-21 and adipocytokines (adiponectin, resistin, leptin and TNF-α) in the NC and the FP group.

**Table 3** Correlation between FGF-21 and other clinical indicators.

| Characteristic | R | 95% CI | *P* value |
|---|---|---|---|
| BMI, kg/m$^2$ | 0.024 | −0.138, 0.197 | 0.797 |
| Leptin, ng/mL | −0.261 | −0.388, −0.121 | 0.005[**] |
| Adiponectin, ug/mL | −0.016 | −0.171, 0.149 | 0.864 |
| Resistin, ng/mL | −0.018 | −0.206, 0.161 | 0.850 |
| TNF-α, pg/mL | 0.010 | −0.172, 0.185 | 0.916 |
| TG, mmol/L | −0.098 | −0.243, 0.098 | 0.297 |
| CHO, mmol/L | −0.136 | −0.277, 0.027 | 0.148 |
| HDL, mmol/L | −0.144 | −0.336, 0.090 | 0.125 |
| LDL, mmol/L | −0.069 | −0.222, 0.108 | 0.463 |
| Glucose, mmol/L | 0.011 | −0.109, 0.189 | 0.908 |

**Notes.**
FGF-21, fibroblast growth factor 21; 95% CI, 95% confidence interval; BMI, body mass index; TNF-α, tumor Necrosis Factor-alpha; TG, triglyceride; CHO, cholesterol; HDL, high density lipoprotein; LDL, low density lipoprotein.
[**]$P < 0.01$

correlation with BMI and serum TG (*Li et al., 2018*). These results suggest that FGF-21 is closely related to metabolic diseases.

The levels of serum FGF-21 in patients with FP, as a common metabolic disease, have not been reported. Johnson et al. showed that FGF-21-deficient mice had fat deposition in pancreatic tissue, which suggests that FGF-21 may be involved in the development of FP (*Johnson et al., 2009*). In this study, serum FGF-21 levels in the FP group were significantly higher than those in healthy controls, which is consistent with the previously

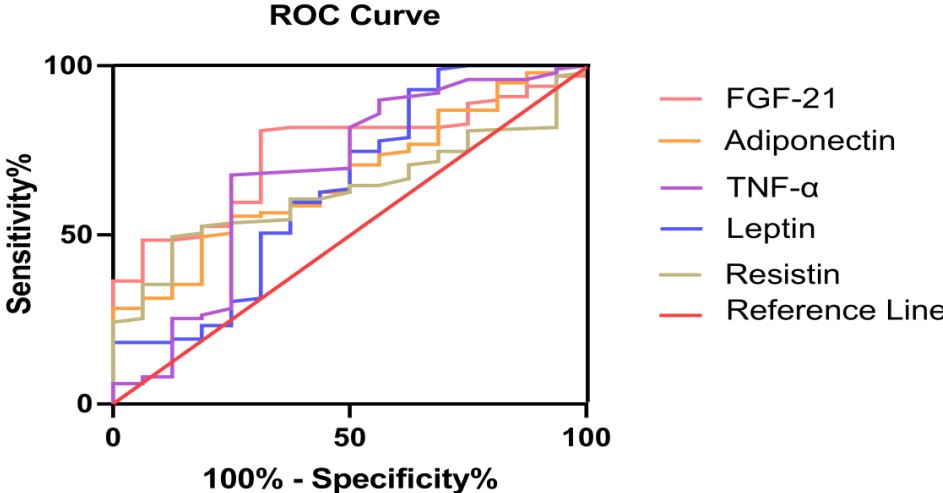

**Figure 2** FGF-21 and adipocytokines (adiponectin, resistin, leptin and TNF-α) for prediction in patients with fatty pancreas: ROC analysis.

**Table 4 Performance of FGF-21 and adipocytokines in predicting fatty pancreas patients.**

| Variables | AUC (95% CI) | Cut-off value | $P$ | Sen, % | Sepc, % |
|---|---|---|---|---|---|
| FGF-21 | 0.744(0.636,0.852) | 171.0 | 0.002[**] | 80.8 | 68.8 |
| Leptin | 0.643(0.478,0.808) | 0.75 | 0.068 | 92.9 | 37.5 |
| Resistin | 0.632(0.518,0.746) | 8.02 | 0.091 | 49.5 | 87.5 |
| Adiponectin | 0.667(0.540,0.793) | 6.72 | 0.033[*] | 49.5 | 81.3 |
| TNF-α | 0.681(0.518,0.844) | 54.33 | 0.020[*] | 67.7 | 75.0 |

**Notes.**
FGF-21, fibroblast growth factor 21;; AUC, area under the curve; Sen, sensitivity; Spec, specificity; 95% CI, 95% confidence intervals; TNF- α, tumor necrosis factor-alpha.
[*]$P < 0.05$
[**]$P < 0.01$

reported changes in serum FGF-21 levels observed in other metabolic diseases, such as fatty liver, obesity, and diabetes (*Li et al., 2018*; *Akour et al., 2017*; *Li et al., 2010*; *Keuper, Häring & Staiger, 2019*). However, this finding does not seem to match the animal experiment of *Johnson et al. (2009)*. The exact reasons are not yet clear. We made possible speculations based on leptin. It is well known that leptin deficiency in mice leads to obesity and insulin resistance (*Sjögren et al., 2019*); however, the current discordant finding is that almost all obese subjects had increased serum leptin levels (*Zhao et al., 2019*). A relatively reasonable explanation is "leptin resistance". It is not clear whether there is "FGF-21 resistance" in populations with metabolic diseases, which requires further study.

We had firstly reported that serum FGF-21 levels were significantly increased in patients with FP and better screened FP patients. In our study, we found that serum FGF-21 levels could better predict FP than leptin, adiponectin, resistin and TNF-α. First of all, as shown in Fig. 1, we intuitively found that FGF-21 had the most obvious increase trend and the statistical difference was also the most significant. Secondly, we also found that the area

under the ROC curve of FGF-21 was more significant and its *P* value was the lowest by observing the prediction of FGF-21 and the other four cytokines on FP patients. ROC curve analysis showed that 171 pg/mL was the best critical value of FGF-21 for predicting FP, and the corresponding sensitivity and specificity are 80.8% and 68.8%, respectively. Hence, serum FGF-21 could better screen patients with FP and be used as a suitable marker for routine clinical use of FP. We preliminarily screened FP patients by detecting the expression levels of serum FGF-21, which would provide some help for clinical prediction of FP. However, whether serum FGF-21 is the outcome indicator of patients with FP needs further follow-up study by our team.

There are several limitations of our study. First, it is difficult to find FP patients without common metabolic syndrome in the clinic. Therefore, the overall sample size of this study was small, and the results may be biased. Second, transabdominal ultrasonography can only be used to identify whether there is fat infiltration in the pancreas, but it cannot quantify the fat content. Therefore, it is difficult to study the correlation between serum FGF-21 and the degree of fat infiltration in the pancreas. Finally, all the serum samples in this study were frozen in a −80 °C refrigerator. The results may deviate from the results obtained from fresh serum samples, but the results are still of reference value.

## CONCLUSION

To sum up, the present study is the first to present the closely relationship between circulating FGF-21 levels and FP patients. Serum FGF-21 levels may be the specific biomarker of FP, which can help identify the population susceptible to FP. The serum FGF-21 lays a clinical foundation for the diagnosis of FP in the future.

**Abbreviations**

| | |
|---|---|
| **FGF-21** | Fibroblast growth factor-21 |
| **FP** | Fatty pancreas |
| **NC** | Normal control |
| **ROC** | Receiver operating characteristic |
| **AUC** | Area under curve |
| **BMI** | Body mass index |
| **ELISA** | Enzyme linked immunosorbent |
| **FBG** | Fasting blood glucose |
| **CHO** | Cholesterol |
| **TG** | Triglyceride |
| **HDL** | High density lipoprotein |
| **LDL** | Low density lipoprotein |
| **TNF-α** | Tumor necrosis factor-alpha |
| **SD** | Standard deviation |
| **UA** | Uric acid |

## ACKNOWLEDGEMENTS

We thank the doctors, nurses, laboratory staff, and study participants for their contributions.

### Funding

This work was supported by the Key Project for Social Development in Jiangsu Province (No. BE2019698), Strengthening Health Care via Science and Education Project and Clinical Medical Innovation Platform Foundation of Yangzhou (No. YXZX20184) and Major public health projects in Yangzhou: Screening projects of early gastrointestinal diseases (2018). The funders had no role in study design, data collection and analysis, decision to publish, or preparation of the manuscript.

### Grant Disclosures

The following grant information was disclosed by the authors:
Key Project for Social Development in Jiangsu Province: BE2019698.
Strengthening Health Care via Science and Education Project and Clinical Medical Innovation Platform Foundation of Yangzhou: YXZX20184.
Major public health projects in Yangzhou: Screening projects of early gastrointestinal diseases (2018).

### Competing Interests

The authors declare there are no competing interests.

### Author Contributions

- Fei Han conceived and designed the experiments, performed the experiments, analyzed the data, prepared figures and/or tables, authored or reviewed drafts of the article, and approved the final draft.
- Ling Yin conceived and designed the experiments, performed the experiments, analyzed the data, prepared figures and/or tables, authored or reviewed drafts of the article, and approved the final draft.
- Xiaoping Yu conceived and designed the experiments, performed the experiments, analyzed the data, prepared figures and/or tables, authored or reviewed drafts of the article, and approved the final draft.
- Renyan Xu performed the experiments, prepared figures and/or tables, and approved the final draft.
- Mingxiang Tian conceived and designed the experiments, prepared figures and/or tables, and approved the final draft.
- Xinnong Liu performed the experiments, prepared figures and/or tables, and approved the final draft.
- Lu Zhou performed the experiments, analyzed the data, prepared figures and/or tables, and approved the final draft.
- Lianghao Hu conceived and designed the experiments, authored or reviewed drafts of the article, and approved the final draft.
- Weijuan Gong conceived and designed the experiments, authored or reviewed drafts of the article, and approved the final draft.

- Weiming Xiao conceived and designed the experiments, authored or reviewed drafts of the article, and approved the final draft.
- Guotao Lu conceived and designed the experiments, authored or reviewed drafts of the article, and approved the final draft.
- Guanghuai Yao conceived and designed the experiments, performed the experiments, authored or reviewed drafts of the article, and approved the final draft.
- Yanbing Ding conceived and designed the experiments, performed the experiments, authored or reviewed drafts of the article, and approved the final draft.

## Human Ethics

The following information was supplied relating to ethical approvals (i.e., approving body and any reference numbers):

The study was approved by Ethics Committee of Affiliated Hospital of Yangzhou University (Ethical Application Ref: 2018-YKL11-27-topic1). All participants agreed to participate in the study, and written informed consent was obtained from each subject.

## Data Deposition

The raw data is available in the Supplementary File.

## Supplemental Information

Supplemental information for this article can be found online at http://dx.doi.org/10.7717/peerj.15176#supplemental-information.

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
