# Peer review of "High circulating fibroblast growth factor-21 levels as a screening marker in fatty pancreas patients"

_PeerJ, doi:10.7717/peerj.15176_

## Round 0.1 · original submission · Minor Revisions

One of the reviewers suggested that the title has little in common with the performed analysis and should be amended.

Reviewer 1 ·

Basic reporting

The manuscript is well written and easy to understand. However, some minor improvement could be made:
Ln 144 – please consider removing word “Moreover”. In my opinion it is unnecessary.
Ln 203-204 – “……hampered” – This sentence is difficult to understand, please rephrase.

The article has sufficient background and context however there are some missing references:
Ln 67 – “… in pancreatic diseases” – citation is needed
Ln 113 – Did authors definite these criteria by themselves? If not, please refer to relevant literature.
Ln 119 – please cite relevant guidelines

Reviewed manuscript has professional structure, was technically written. Hypothesis is self-contained. Results are relevant to the current state of knowledge and literature.

Experimental design

Research question was well defined, consistent with the existing literature. Fills some gap in knowledge. Very high technical standard of the analysis however, I found following issues that need to be clarified:
Ln 130 – please clarify, what tests did author use?
Ln 143 – Was it statistically significant? Please report P value
Ln 144 – please report P value

This investigation has good technical standard. Methods were described with sufficient details to replicate. Good technical standard. Meets ethical requirements.

Validity of the findings

The main problem with the study, that authors did not compare examined cytokines between each other. There could be an additional paragraph in Discussion section, that would answer following questions:
• Is FGF-21 better indicator of fatty pancreatitis than leptin, adiponectin, and TNF-a?
• What marker did exhibit better parameters for screening or final diagnosis of FP?
• Is FGF-21 a suitable marker for routine clinical use? How would you use it?
• Is there any potential for patient’s outcome indicator?
Please discuss and make relevant conclusions. In current form, Conclusions section did not correspond to the amount of analysis performed. It should be significantly expanded (at least 2-3 sentences).
Secondly, the title does not correspond to the manuscript content and the analysis. Where did you evaluate the patient prognosis in context of elevated FGF-21? Please change title if possible.
I do not have access to the IBM SPSS statistics; thus, I did not recalculate the values. However, it statistically sound and data seemed to be controlled.

Additional comments

Good technical study, suitable for publication. Need some improvements. Accept after minor revisions.

·

Basic reporting

The manuscript is well written and literature/background is sufficient and supports the findings.

Experimental design

Present study is well designed but following areas require some attention.
What is the purpose of determining uric acid and TNF in the present study? Please mention its possible relation with the study design and findings of the study.
Procedure of analysis of some parameter such as Uric acid, TNF is not explained.
Results of uric acid levels are not even mentioned in the results section.
Inclusion and exclusion criteria for NC (normal control) is not explained.
Patients gender is not mentioned in the methodology section.

Validity of the findings

Conclusion is not well written. It should be properly aligned with the findings of the study.
Some of the findings are not discussed nor mentioned in the results section, they are only present in tables. It is suggested to include them in results section and also discussed properly.
What is the possible mechanism by which leptin levels can affect FP. Please elaborate.
Is there any significant role that smoking and drinking can play in the underlying pathological mechanism of FP.

Additional comments

The manuscript submitted 80033v1 explores the role of serum levels of fibroblast growth factor-21 (FGF-21) in fatty pancreas (FP) patients and to investigate their potential clinical value. Authors illustrate that serum FGF-21 levels are closely related to FP. The detection of serum FGF-21 levels may help identify the population susceptible to FP. Body mass index, fasting blood glucose levels, uric acid levels, cholesterol levels, high-density lipoprotein level, levels of serum FGF-21, resistin, leptin, tumor necrosis factor-α were analyzed in the present study. This study will help for earlier diagnosis/detection of FP based on serum FGF-21 levels. The study is well designed and the findings are interesting and have scientific values, but the manuscript should undergo minor revision to meet the standard of the journal

Reviewer 3 ·

Basic reporting

Overall, the manuscript is well-written. The tables and figures are clear. Minor points regarding the references cited in the Introduction, two related publications of FGF21 in the pancreas are missing, it would be helpful to add (PMID: 28089565 and PMID: 31915301).

Experimental design

Original primary research suits the journal. The research question of the current study is well-defined. Methods are clearly described.

Minor comments: it would be helpful to describe how those fatty pancreas patients are identified. Do they have any symptoms? Are there any control who have the equivalent BMI, fatty liver, diabetes, obesity, hypertension and dyslipidemia, but do not have fatty pancreas? Are those parameters, including BMI, fatty liver, diabetes, obesity, hypertension and dyslipidemia, independent risk factors of fatty pancreas?

Validity of the findings

Findings are interesting and useful. Conclusions are well-stated. One out-of-interest question is whether the levels of FGF21 or adipocytokines differ between male and females of fatty pancreas patients group.

Additional comments

No

---

## Round 0.2 · accepted · Accept

The authors made changes according to the reviewer's suggestions.